# Timing of Medium-Chain Triglyceride Consumption Modulates Effects in Mice with Obesity Induced by a High-Fat High-Sucrose Diet

**DOI:** 10.3390/nu14235096

**Published:** 2022-12-01

**Authors:** Tomoki Abe

**Affiliations:** Healthy Food Science Research Group, Cellular and Molecular Biotechnology Research Institute, National Institute of Advanced Industrial Science and Technology (AIST), Tsukuba 305-8566, Japan; tomoki-abe@aist.go.jp; Tel.: +81-29-861-6053

**Keywords:** medium chain triglyceride, circadian rhythm, obesity, lipolysis, hormone-sensitive lipase

## Abstract

The prevalence of obesity is increasing worldwide, and obesity can cause type 2 diabetes, atherosclerosis, hypertension, cardiovascular disease, and cancer. Intake of medium-chain triglycerides (MCTs) containing medium-chain fatty acids reduces body fat and insulin resistance in rodents and humans. This study aimed to determine how the timing of MCT consumption affects obesity and metabolic dysfunction induced in mice by a high-fat high-sucrose diet (HFHSD). Mice received an HFHSD with or without MCT (M-HFHSD) during either the active or rest phase for 9 weeks. Significant reduction in body weight, white adipose tissue (WAT) weight, and adipocyte size in epididymal WAT (eWAT) and improved insulin sensitivity in mice fed with M-HFHSD during the active but not the rest phase were observed. The consumption of M-HFHSD during both active and rest phases increased glucose tolerance. Phosphorylated Akt was more abundant in the gastrocnemius muscles and eWAT of M-HFHSD-fed mice than in those fed HFHSD during the active phase. The mRNA and protein expression of lipogenic genes increased in the eWAT of mice fed M-HFHSD compared with those fed HFHSD. Feeding with M-HFHSD during the active phase significantly increased the abundance of phosphorylated Ser563 and 660 of hormone-sensitive lipase and its upstream protein kinase A in eWAT. These results indicated that the timing of consumption modulates the effects of MCT on eWAT hypertrophy and glucose and lipid metabolism in mice.

## 1. Introduction

The prevalence of obesity is increasing worldwide and is associated with type 2 diabetes, atherosclerosis, hypertension, cardiovascular diseases, and cancer [1]. Nutritional excess is a major contributor to obesity and causes hypertrophy of white adipose tissue (WAT) and the accumulation of ectopic fat in the liver, skeletal muscle, and kidneys. Overabundance of visceral and ectopic fat is associated with the development of insulin resistance [1]. Therapeutic strategies are required to reduce fat, improve insulin resistance, and enhance metabolic function.

Time-restricted feeding (TRF) is a type of intermittent fasting used as a nutritional intervention strategy to induce weight loss. Overweight or obese adults can safely adopt an 8–10 h interval of TRF to facilitate weight loss [2,3,4]. While TRF during the active phase confers many health benefits, TRF during the rest phase adversely affects metabolism. Arble et al. [5] reported that mice gain more weight when fed only during the 12 h-rest, than during the 12 h-active phase. In this study, it was found that TRF results in physical inactivity and muscle atrophy in mice fed a high-fat high-sucrose diet (HFHSD) during rest compared with the active phase [6]. These results indicate that feeding time is associated with body fat accumulation and metabolic dysfunction.

The time of day when nutrients are consumed is vital for maximizing health benefits. In mice fed with high-fructose diet and humans, fish oil containing docosahexaenoic and eicosapentaenoic acids prevents hyperlipidemia more effectively when consumed in the morning than in the evening [7,8]. Aoyama et al. [9] showed that protein induces more hypertrophic effects in mouse skeletal muscle when consumed during the early than the late active phase, depending on the muscle clock. The skeletal muscle index and hand grip strength increase more in elderly people who consume protein during breakfast than during dinner [9]. The flavonoid catechin reduces body fat and improves blood glucose levels in patients with type 2 diabetes [10,11]. Catechin suppresses postprandial blood glucose in humans more effectively when ingested in the evening than in the morning [12].

Medium-chain triglycerides (MCTs) comprise medium-chain fatty acids (MCFAs) containing 6–12 carbon chains and glycerol and induce less weight gain than long-chain fatty acids (LCFAs) [13]. Adiposity is reduced and insulin sensitivity is preserved in the adipose tissues and skeletal muscles of mice fed MCFAs compared with those fed LCFAs [14]. Dietary MCFAs prevent fat accumulation mediated by increased expression of genes associated with energy metabolism in the abdominal subcutaneous adipose tissues of abdominally obese humans [15]. In addition, compared with long-chain triglycerides (LCTs), MCT supplementation increases lipolysis through the phosphorylation of hormone-sensitive lipase (HSL) and protein kinase A (PKA) in the WAT of mice [16,17]. Abe et al. [18] recently examined the effects of the time of daily MCT supplementation on muscle mass, function, and cognition. Although supplementing elderly residents of a nursing home with MCT significantly increased muscle mass and function when consumed at breakfast and dinnertime [18], the effects of the time of dietary MCT consumption on body fat and insulin sensitivity remain unclear.

This study aimed to determine the effects of the time of day of MCT consumption on body fat and glucose and lipid metabolism in mice with diet-induced obesity. MCT reduced WAT weight and ameliorated insulin resistance induced by HFHSD when consumed during the active phase compared to the resting phase in mice. The time of MCT intake affects the expression of lipogenic genes in the liver and epididymal WAT (eWAT). Dietary MCT consumption during the active phase, but not the rest phase, increased HSL phosphorylation, which is associated with lipolytic activity in eWAT in mice. The findings of this study indicated that MCT induces significant anti-obesogenic effects in mice when consumed during the active phase compared to the rest phase.

## 2. Materials and Methods

### 2.1. Animals and Study Design

Five-week-old male wild-type C57BL/6J mice (Japan SLC Inc., Shizuoka, Japan) were housed in cages with free access to CE2 normal chow (CLEA Japan, Inc., Tokyo, Japan) for 3 weeks under a 12 h light-12 h dark cycle (LD 12:12, lights on at Zeitgeber time (ZT) 0 and lights off at ZT12) and a constant temperature of 22 ± 1 °C. Eighty mice in total were used for this study. Eight-week-old mice had access to food ad libitum (*n* = 16) or a high-fat high-sucrose diet (HFHSD; Oriental Yeast Co. Ltd., Tokyo, Japan) or HFHSD containing 13% medium-chain triglyceride (MCT; ratio of C8:0 to C10:0 = 1:3; Nisshin Oillio Group, Tokyo, Japan) (M-HFHSD) only during the active (APF; ZT14-22, *n* = 32) or rest (RPF; ZT2-10, *n* = 32) phase for 9 weeks (Appendix A). Table 1 shows the composition of HFHSD and M-HFHSD. The control diet (HFHSD) contained 7% beef tallow and 6% lard, instead of MCT. Seventeen-week-old mice were sacrificed at ZT2, 8, 14, and 20, and tissues were dissected, weighed, and rapidly frozen in liquid nitrogen. The wheel-running activity of six mice in each group was measured. Collection of tissues and blood was done at ZT2, 8, 14, and 20 (four mice at each time point).

### 2.2. Quantitation of Blood Hormones and Metabolic Parameters

Blood obtained at ZT2, 8, 14, and 20 was immediately separated by centrifugation for 20 min at 3000× *g*, and then, serum was stored at −80 °C. Serum glucose, non-esterified fatty acid (NEFA), triglyceride, total cholesterol, β-hydroxybutyrate, and glycerol concentrations were measured using LabAssay™ Glucose, NEFA, Triglyceride, and Cholesterol (FUJIFILM Wako Pure Chemical Corporation, Tokyo, Japan), β-Hydroxybutyrate Colorimetric and Free Glycerol Colorimetric/Fluorometric Assay Kits (BioVision Inc., Milpitas, CA, USA), respectively. Serum concentrations of insulin and leptin were measured using Ultra Sensitive Mouse Insulin and Leptin ELISA kits (Morinaga Institute of Biological Science, Kanagawa, Japan), respectively.

### 2.3. Tissue Histology

Epididymal and inguinal subcutaneous WAT was fixed in 10% formalin neutral buffer (FUJIFILM Wako Pure Chemical Corporation, Tokyo, Japan) for 24 h at 4 °C and embedded in paraffin following the standard protocol. Sections (5 μm) were stained with hematoxylin and eosin for 10 and 5 min at room temperature, respectively, and at least 750 adipocytes in eWAT from each mouse were visualized using a BZ-X810 microscope (Keyence, Osaka, Japan) and enumerated using a hybrid cell count software (BZ-X800 Analyzer, ver. 1.1.2.4, Keyence).

### 2.4. Glucose and Insulin Tolerance Tests

Glucose and insulin tolerance were evaluated at ZT8 and 20 in mice after 8 weeks of HFHSD or M-HFHSD (Appendix A). Mice were subjected to fasting for 4 h with free access to water during the feeding phase, then administered intraperitoneally (i.p.) with 1.5 g/kg glucose or 1 U/kg insulin (Sigma-Aldrich Corp., St. Louis, MO, USA) at ZT8 (mice fed during the rest phase) or at ZT20 (fed during the active phase). The mice had access to water ad libitum for 10 h while fasting and were then administered i.p. with 1.5 g/kg glucose or 1 U/kg insulin at ZT8 (mice fed during the active phase) or ZT20 (fed during the rest phase). Blood glucose levels were measured at 0, 15, 30, 60, and 120 min using an Accu-Chek Aviva Nano kit (Roche Diagnostics, Mannheim, Germany).

### 2.5. Real-Time Reverse Transcription Quantitative Polymerase Chain Reaction

Total RNA was isolated from mouse livers, eWAT, and gastrocnemius muscles using RNAiso Plus (Takara Bio Inc., Shiga, Japan), and cDNA was synthesized using PrimeScript™ RT reagent kits with a gDNA eraser (Takara Bio). Real-time reverse transcription quantitative polymerase chain reaction (RT-qPCR) was performed using SYBR^®^ Premix Ex TaqTM II (Takara Bio) and a LightCycler™ (Roche), with the primers shown in Appendix A. The amplification conditions were as follows: 95 °C for 10 s, followed by 45 cycles at 95 °C for 5 s, 57 °C for 10 s, and 72 °C for 10 s. The internal controls were glyceraldehyde-3-phosphate dehydrogenase (GAPDH) and 36b4.

### 2.6. Western Blotting

Mouse livers, WAT, and gastrocnemius muscle were homogenized in RIPA buffer (FUJIFILM Wako Pure Chemical Corporation, Tokyo, Japan) containing protease (cOmplete, Roche) and phosphatase inhibitor tablets (PhosSTOP, Sigma-Aldrich Corp.), and protein extracts (10 μg/lane) were resolved by 8% SDS or 10% SDS-PAGE and transferred to polyvinylidene difluoride membranes (Bio-Rad). Non-specific antigen binding on the membranes was blocked by incubation with 4% Block Ace (DS Pharma Biomedical Co., Ltd., Osaka, Japan) for 1 h, followed by incubation with primary antibodies at 4 °C overnight. The details pertaining to primary antibodies procured from: suppliers are as follows: anti-acetyl-CoA carboxylase (ACC; catalog no.: 3676), anti-Akt (catalog no. 2920), anti-phospho-Akt (Ser473; catalog no.: 9271), anti-phospho-Akt (Thr307; catalog no.: 9275), anti- extracellular signal-regulated kinase (ERK) 1/2 (catalog no: 4695), anti-phospho-ERK1/2 (Thr202/Tyr204; catalog no: 4370), anti-fatty acid synthase (FAS; catalog no: 3180), anti-HSL (catalog no: 18381), anti-phospho-HSL (Ser563; catalog no: 4139), anti-phospho-HSL (Ser660; catalog no: 45804), anti-phospho-protein kinase A (PKA) substrate (catalog no: 9624), anti-peroxisome proliferator-activated receptor gamma (PPAR; catalog no: 2435; Cell Signaling Technology, Danvers, MA, USA) and anti-GAPDH (catalog no. NB300-221; Novus Biologicals, LLC, Centennial, CO, USA). Proteins were detected using an enhanced chemiluminescence system (Wako Pure Chemical Industries). Proteins were quantified using ImageJ software (ver. 1.53f, https://imagej.nih.gov/ij/, accessed on 22 February 2022).

### 2.7. Statistical Analysis

All data are expressed as mean ± SEM and were statistically assessed using Excel-Toukei 2010 (Social Survey Research Information Co. Ltd., Osaka, Japan). Student’s *t*-test or two-way analysis of variance (ANOVA) test was followed by Tukey-Kramer multiple comparison tests. Values with *p* < 0.05 were considered significant.

## 3. Results

### 3.1. Dietary MCT Did Not Affect Voluntary Wheel-Running

The mice were given an HFHSD containing MCT only during the active or rest phase for 8 h per day to determine whether the timing of consumption altered diet-induced obesity and voluntary wheel-running activity. Nocturnal wheel-running rhythms were maintained in mice fed during the active and rest phases and in mice fed ad libitum (Figure 1A). The consumption of MCT did not change the total daily wheel-running activities of the mice fed during the active phase and those fed ad libitum. Hourly counts of wheel-running at ZT12 were 36% lower in mice fed M-HFHSD than in those fed HFHSD during the rest phase (Figure 1A). Eight weeks of M-HFHSD did not affect the circadian rhythms of wheel-running in mice fed ad libitum or during the active phase. Daily wheel-running activity on day 0 was 27% lower in mice given M-HFHSD than in those given HFHSD during the rest phase (Figure 1B).

### 3.2. Effects of Dietary MCT during the Active or Resting Phases on HFHSD-Induced Fat Accumulation in Mice

Consistent with previous findings, the mice weighed 11% less when fed ad libitum with M-HFHSD than with HFHSD (Figure 2A). MCT consumption only during the active phase significantly reduced HFHSD-induced body weight gain in the mice (Figure 2A). However, intake of MCT only during the rest phase did not suppress HFHSD-induced body weight gain in mice (Figure 2A). Supplementation with MCT did not change food intake in mice fed ad libitum or during the active or rest phase. (Figure 2B). The weight of the liver and gastrocnemius muscles did not differ significantly between the HFHSD and M-HFHSD groups fed ad libitum or during the active or rest phase (Figure 2C,D). The eWAT and subcutaneous WAT (sWAT) weighed relatively less in mice fed an M-HFHSD than HFHSD ad libitum or during the active phase (Figure 2E,F). Figure 2G shows the areas of adipocytes in the eWAT of mice fed ad libitum or during the active and resting phases. The average adipocyte area was significantly smaller in mice fed M-HFHSD than in those fed HFHSD during the active phase (Figure 2H). The size of the average area of adipocytes did not significantly differ between the HFHSD and M-HFHSD groups fed ad libitum or during the rest phase (Figure 2H).

### 3.3. Effects of MCT Consumed during the Active and Rest Phases on Serum Metabolic Parameters in Mice

Serum levels of insulin, leptin, glucose, β-hydroxybutyrate, and NEFA did not significantly differ between mice fed with HFHSD and M-HFHSD during the active and rest phases (Figure 3A–E). Serum free glycerol values at ZT8 were 19% lower in mice fed M-HFHSD than in those fed HFHSD during the rest phase (Figure 3F). Serum triglyceride values at ZT2 were significantly higher in mice fed M-HFHSD than in those fed HFHSD during the active and rest phases, respectively (Figure 3G). Serum total cholesterol values at ZT2 were 64% higher in mice fed M-HFHSD than in those fed HFHSD during the rest phase (Figure 3H).

### 3.4. Effects of MCT Consumed during Active or Rest Phases on Glucose Tolerance and Insulin Sensitivity in Mice Fed HFHSD

As shown in Figure 4A, the two-way ANOVA test revealed a significant and important effect of diet and time. However, the two-way ANOVA test did not show a significant difference at each time point. MCT consumption led to increased insulin sensitivity in mice fed with HFHSD during the active phase but not the rest phase (Figure 4A). In contrast, insulin sensitivity during fasting was exacerbated in mice fed M-HFHSD compared to those fed HFHSD during the active phase (Figure 4B). Consumption of MCT did not change insulin sensitivity, regardless of feeding or fasting in mice fed with HFHSD during the rest phase. An intraperitoneal glucose challenge revealed improved glucose tolerance in mice fed M-HFHSD compared to those fed HFHSD at 60 min after glucose injection during the active phase (Figure 4C). However, glucose tolerance was impaired 15 min after glucose injection in mice fed M-HFHSD compared to those fed HFHSD during the rest phase. MCT consumption improved glucose tolerance during fasting in mice fed M-HFHSD compared with mice fed HFHSD during the rest phase (Figure 4D). The amount of Akt phosphorylation at Ser473 or Thr308 at ZT8 and 20 in the liver, gastrocnemius muscles, and eWAT of mice that were fed during the active and rest periods was measured. In the livers of mice that were fed during the active and rest phases, the levels of Akt phosphorylation on Ser473 and Thr308 were comparable between HFHSD and M-HFHSD (Figure 4E). The amount of phosphorylation at Ser473 of Akt was higher by 72% at ZT20 in the gastrocnemius muscles of mice fed M-HFHSD than in those fed HFHSD during the active phase (Figure 4F). MCT consumption during the rest phase did not change Ser473 and Thr308 phosphorylation of Akt in the gastrocnemius muscles of mice fed with HFHSD. During the active phase, the amount of phosphorylation at Thr308 of Akt at ZT8 was higher by 47% in the eWAT of mice fed with M-HFHSD than those fed with HFHSD (Figure 4G), whereas phosphorylation at Ser473 and Thr308 in eWAT at ZT8 and ZT20 was similar between mice fed with M-HFHSD and HFHSD during the rest phase.

### 3.5. Expression of Genes and Proteins Associated with Lipid Metabolism in the Liver of Mice Fed with HFHSD or M-HFHSD

To elucidate the molecular mechanisms underlying the suppression of fat accumulation by MCT consumed during the active phase, the expression of genes associated with fatty acid oxidation in the gastrocnemius muscle of mice was assessed. Notably, dietary MCT did not increase the mRNA expression of *carnitine palmitoyltransferase 1b* (*Cpt1b*), *pyruvate dehydrogenase kinase 4* (*Pdk4*), *peroxisome proliferator-activated receptor (PPAR)-γ coactivator-1α* (*Pgc-1α*), and *Pparα* in the gastrocnemius muscles of mice fed with HFHSD during the active and rest phases (Appendix A). Therefore, the expression of lipogenic genes in the liver of mice in the active and rest phases was assessed. More *fatty acid synthase* (*Fasn*) and *acetyl-CoA carboxylase 1* (*Acc1*) were expressed at ZT2 in the liver of mice fed M-HFHSD compared to those fed HFHSD during the active phase (Figure 5A,B). MCTs did not affect the mRNA levels of *sterol regulatory element-binding protein 1c* (*Srebp1c*) and *Pparγ1* in the liver of mice fed HFHSD during the active period (Figure 5C,D). Dietary MCT did not affect the hepatic mRNA levels of *Fasn*, *Acc1*, *Srebp1c*, and *Pparγ1* in mice fed with HFHSD during the rest phase (Figure 5A–D). The hepatic protein levels of lipogenesis-related enzymes FAS, ACC, and PPARγ1 was assessed by Western blotting (Figure 5E). MCT consumption reduced hepatic FAS at ZT8 in mice during the rest phase, but not in the active phase (Figure 5F). Dietary MCT did not change the abundance of hepatic ACC in mice fed HFHSD during the rest or active phase (Figure 5G). The abundance of PPARγ1 was 33% lower at ZT20 in the liver of mice fed M-HFHSD than in those fed HFHSD during the active phase (Figure 5H). MCT consumption did not affect hepatic Pparγ1 abundance in mice that were fed during the active period.

### 3.6. Expression of Genes and Proteins Associated with Lipid Metabolism in eWAT of Mice Fed with HFHSD or M-HFHSD

Increased *Fasn* was expressed at ZT2 in the eWAT of mice fed with M-HFHSD than in those fed with HFHSD during the active period (Figure 6A). Dietary MCT increased the mRNA expression of *Fasn* and *Acc1* at ZT14 in the eWAT of mice fed during the rest period (Figure 6A,B). Less *Srebp1c* was expressed at ZT20 in the eWAT of mice fed with M-HFHSD than in those fed with HFHSD during the rest period (Figure 6C). MCT consumption did not affect *Pparγ2* expression in the eWAT of mice fed with HFHSD during the rest phase (Figure 6D). The consumption of MCT did not affect the mRNA expression of *Acc1*, *Srebp1c*, and *Pparγ2* in eWAT of mice fed with HFHSD during the rest phase (Figure 6B–D). The protein levels of lipogenesis-related enzymes FAS, ACC, and PPARγ1/2 in eWAT from mice fed during the rest and active periods were assessed using Western blotting (Figure 6E). The eWAT of mice contained more FAS, ACC, and PPARγ1/2 proteins when the mice were fed with M-HFHSD than those fed with HFHSD during the active phase (Figure 6F–H). Dietary MCT increased the protein levels of FAS and ACC at ZT8 in the eWAT of mice fed with HFHSD during the rest period (Figure 6F,G).

### 3.7. Expression of Genes and Proteins Associated with Lipolysis in eWAT of Mice Fed with HFHSD or M-HFHSD

Dietary MCT did not affect the mRNA expression of *Atgl* and *Hsl* in eWAT of mice fed with HFHSD during the active period (Figure 7A,B). Less *Atgl* was expressed at ZT20 in the eWAT of mice fed with M-HFHSD than in those fed with HFHSD during the rest period (Figure 7A). Consumption of MCT did not affect the mRNA expression of *Hsl* in the eWAT of mice fed with HFHSD during the rest period (Figure 7B). Western blotting was used to assess the protein levels of the lipolysis-related enzymes ATGL, phosphorylated HSL at Ser563, phosphorylated HSL at Ser660, and total HSL in eWAT from mice fed during active rest periods (Figure 7C). Dietary MCT did not affect the abundance of ATGL in eWAT of mice fed with HFHSD during the active period (Figure 7D). The abundance of ATGL at ZT8 was higher by 41% in the eWAT of mice fed M-HFHSD than in those fed with HFHSD during the rest period (Figure 7D). Consumption of MCT increased the phosphorylation of HSL at Ser563 and Ser660 at ZT8 in the eWAT of mice fed with HFHSD during the active period (Figure 7E,F). In contrast, dietary MCT did not affect the abundance of phosphorylated HSL at Ser563 and Ser660 in the eWAT of mice fed with HFHSD during the rest period. The abundance of phosphorylated HSL at Ser660 in mice fed ad libitum did not differ between the diets (Appendix A). Phosphorylation of HSL at residues Ser563 and Ser660 is induced by PKA [19,20]. Extracellular regulated kinase1/2 modulates lipolysis by regulating HSL activity [21]. Western blotting was used to assess the protein levels of the phosphorylated PKA substrates, phosphorylated extracellular signal-regulated kinase 1/2 (ERK1/2), and total ERK1/2 in the eWAT of mice fed during the active and rest periods (Figure 7G). All images of western blotting used for densitometric analysis were shown in Appendix A. The amounts of phosphorylated PKA substrates and ERK1/2 at ZT8 were increased by 92% and 96%, respectively, in eWAT from mice fed with the M-HFHSD than HFHSD during the active phase (Figure 7H,I). However, MCT consumption during the rest phase did not change the amount of phosphorylated PKA substrates and phosphorylated ERK1/2.

## 4. Discussion

Obesity and metabolic dysfunction can be prevented by MCT in rodents and humans [13,14,15]. Abe et al. [18] recently examined the effects of the time of MCT supplementation on muscle mass and function in elderly residents of a nursing home. In this study, the effects of dietary MCT during active and rest phases on body fat accumulation and glucose and lipid metabolism in mice were compared; MCT during the active phase, but not during the rest phase, reduced body weight gain and WAT hypertrophy induced by HFHSD in mice without changes in physical activity and food consumption. Insulin sensitivity and glucose tolerance were improved in mice fed with HFHSD during the active phase but not the rest phase. Furthermore, MCT intake during the active and rest phases increased the expression of lipogenic genes in the eWAT of mice fed with HFHSD. The abundance of phosphorylated HSL at Ser563 and Ser660 was higher in mice fed with M-HFHSD than in those fed with HFHSD during the active period. However, these increases were not observed in the eWAT of mice fed during the rest phase. These results suggest that the time of dietary MCT consumption modulates the effects of HFHSD on body fat accumulation and glucose and lipid metabolism in mice.

Time-restricted feeding (TRF) is a potent therapy for obesity. TRF suppresses diet-induced body weight gain and metabolic dysfunction in mice only when administered during the active phase [22]. In this study, the effects of TRF and voluntary wheel-running to quantify the physical activity and circadian rhythms of wheel-running activity were examined. Notably, dietary MCT did not change the daily counts and circadian rhythms of wheel running in mice fed with HFHSD. Wilkinson et al. [23] showed that the benefits of statin and antihypertensive therapy were additive to the effects of TRF in humans without any changes in physical activity. Consistent with this finding, dietary MCT enhanced the suppression of diet-induced obesity and metabolic dysfunction by TRF only when consumed during the active phase without an increase in voluntary wheel-running activity.

The objective of this study was to elucidate the effects of the time of day of MCT consumption on lipid metabolism in mice. There were no significant differences in liver weight between HFHSD and M-HFHSD (Figure 2C). In contrast, MCT intake affected the serum TG levels in mice (Figure 3G). Hepatic lipogenesis produces triglycerides from fatty acids, which are secreted into the blood, resulting in increased serum triglyceride concentrations. Therefore, the mRNA and protein expression levels of lipogenesis-related genes in the liver of the mice was measured. The intake of MCT during the active phase increased the mRNA expression levels of *Fasn* and *Acc1* at ZT2 and decreased PPARγ1 protein level at ZT20 (Figure 5A,B,H). In contrast, MCT intake during the rest phase did not change the expression levels of lipogenic genes and decreased the protein levels of FAS at ZT8. These results indicated that the time of MCT intake may modulate hepatic lipid metabolism in mice.

MCT consumption during the active phase significantly reduced eWAT weight and adipocyte size in mice fed with HFHSD. The amounts of phosphorylated HSL on Ser563 and Ser660, which are associated with lipolytic activity [19], were increased in eWAT from mice fed with M-HFHSD compared to those fed HFHSD, during the active period. In contrast, dietary MCT consumption during the active phase increased the protein expression of lipogenic genes such as FAS, ACC, and PPARγ2 in eWAT, suggesting increased adipose lipogenic activity. Capric acid (C10:0) binds to and partially activates *PPARγ* without inducing adipogenesis in 3T3-L1 cells [24,25]. SREBP1c is a major transcription factor that regulates lipogenesis-associated genes [26]. SREBP1c expression is regulated by insulin. However, MCT intake only during the active phase did not change the concentrations of serum insulin (Figure 3A) and mRNA expression levels of *Srebp1c* in WAT (Figure 6C). PPARγ regulates the expression of lipogenesis-related genes, such as *Fasn* and *Acc* [27,28], which indicates that increased PPARγ is associated with increased expression of FAS and ACC in WATs of mice fed M-HFHSD compared with those of mice fed HFHSD. These findings suggest that despite higher adipose lipogenic activity, lipolysis and lipid consumption might be enhanced in mice fed with M-HFHSD than in those fed with HFHSD during the active phase, resulting in suppressed diet-induced WAT hypertrophy.

In this study, it was found that dietary MCT did not increase the mRNA expression of genes associated with fatty acid oxidation in the mouse gastrocnemius muscles. Medium-chain FAs induce the expression of mitochondrial proteins associated with FA oxidation in murine skeletal muscles and myocytes [14,29]. Lundsgaard et al. [30] recently showed that dietary MCFAs rescue insulin action and increase basal fatty acid oxidation in humans fed with a high-fat diet (HFD). Further studies are needed to confirm the effects of MCT intake time on fatty acid oxidation.

In this study, dietary MCT did not affect the serum levels of insulin and glucose in mice fed with HFHSD. Insulin tolerance tests revealed that dietary MCT consumed during the active phase moderately improved the systemic insulin sensitivity in mice. Glucose tolerance was improved in mice fed with M-HFHSD compared to those fed with HFHSD. The abundance of phosphorylated Akt, a key insulin signaling molecule, was increased in the gastrocnemius muscles and eWAT (without changes in serum insulin levels), but not in the liver of mice fed with M-HFHSD during the active phase. These findings suggest that dietary MCT selectively improves insulin sensitivity in mouse gastrocnemius muscles and eWAT.

Ingested meals stimulate the intestinal secretion of incretins such as glucagon-like peptide-1 (GLP-1) and gastric inhibitory polypeptide/glucose-dependent insulinotropic polypeptide (GIP) [31]. For the reason that GLP-1 increases insulin release from pancreatic β-cells in a glucose-dependent manner, incretins are important for the compensatory release of insulin in obesity [32]. GIP promotes energy storage mediated by lipogenesis in adipose tissue. Murata et al. [33,34] found that dietary MCT prevents diet-induced obesity by inhibiting LCT-induced GIP secretion in mice. Ingesting MCTs containing only capric acid increases plasma GLP-1 levels that suppress obesity and insulin resistance induced by HFD in mice [35]. Daily feeding and fasting cycles affect fluctuations in blood GLP-1 and GIP [36,37]. Further study will be carried out in the near future in order to elucidate the effects of dietary MCT timing on the release of intestinal GLP-1 and GIP.

Supplementation with MCT increases the abundance of phosphorylated HSL in WAT of mice [16,17]. Consistent with these findings, dietary MCT increased the levels of phosphorylated HSL at Ser563 and Ser660 and its upstream PKA activity at ZT8 in WAT of mice fed during the active phase. However, MCT consumption did not affect the abundance of phosphorylated HSL at Ser563 and Ser660 in WAT of mice fed ad libitum or during the rest phase. Further studies are needed to elucidate the molecular mechanisms underlying MCT-induced HSL phosphorylation during the active phase.

## Figures and Tables

**Figure 1 nutrients-14-05096-f001:**
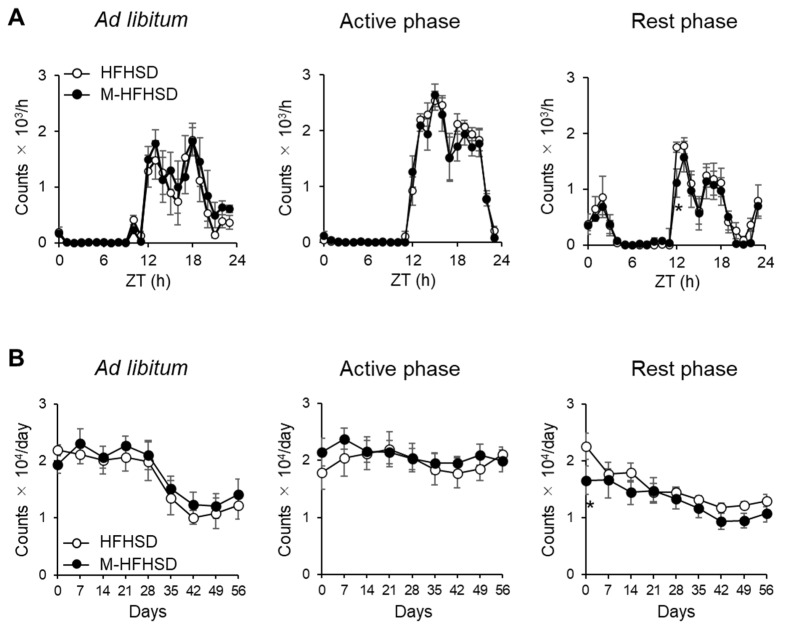
Effect of MCT intake on voluntary wheel-running activity in mice. (**A**) Hourly count/24 h of mouse wheel-running activity after 8 weeks of consuming HFHSD or M-HFSHD) (**B**) Daily wheel-running activity by mice fed with HFHSD or M-HFHSD. Data are shown as means ± SEM (*n* = 6). * *p* < 0.05 vs. M-HFHSD at the same time. HFHSD, high-fat high-sucrose diet; MCT, medium-chain triglyceride; M-HFHSD, high-fat high-sucrose diet containing medium-chain triglyceride; ZT, zeitgeber time.

**Figure 2 nutrients-14-05096-f002:**
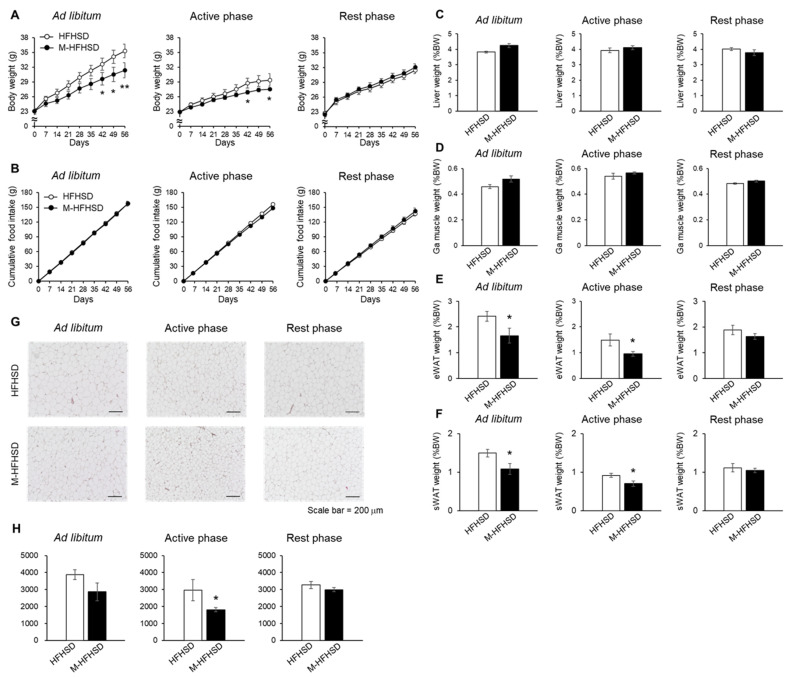
Effects of MCT on body weight, food intake, and tissue weight. (**A**) Time course of changes in BW. (**B**) Average cumulative food intake by mice. Relative weights of the liver (**C**), Ga (**D**), eWAT (**E**), and sWAT (**F**) to BW. (**G**) Representative HE stained images of mouse eWAT. Scale bar = 200 μm. (**H**) Average area of adipocytes in eWAT from mice. Data were collected after feeding with HFHSD or M-HFSHD for 9 weeks. Data are shown as mean ± SEM (*n* = 8). * *p* < 0.05 and ** *p* < 0.01 vs. HFHSD. BW, body weight; Ga, gastrocnemius muscle; HFHSD, high-fat high-sucrose diet; MCT, medium-chain triglyceride; M-HFHSD, high-fat high-sucrose diet containing medium-chain triglyceride; eWAT, epididymal white adipose tissue; sWAT, inguinal subcutaneous white adipose tissue; HE, hematoxylin and eosin.

**Figure 3 nutrients-14-05096-f003:**
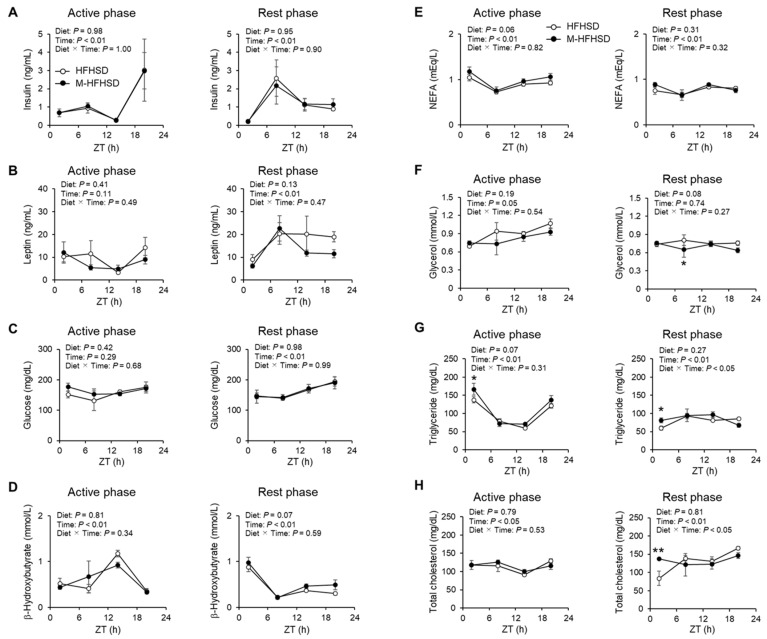
Serum hormone levels and metabolic parameters. Serum levels of insulin (**A**), leptin (**B**), glucose (**C**), β-hydroxybutyrate (**D**), NEFA (**E**), glycerol (**F**), triglyceride (**G**), and total cholesterol (**H**) at indicated times in mice fed with HFHSD or M-HFHSD for 9 weeks. Data are shown as mean ± SEM (*n* = 4). * *p* < 0.05 and ** *p* < 0.01 vs. HFHSD at same time. HFHSD, high-fat high-sucrose diet; MCT, medium-chain triglyceride; M-HFHSD, high-fat high-sucrose diet containing medium-chain triglyceride; NEFA, non-esterified fatty acid; ZT, zeitgeber time.

**Figure 4 nutrients-14-05096-f004:**
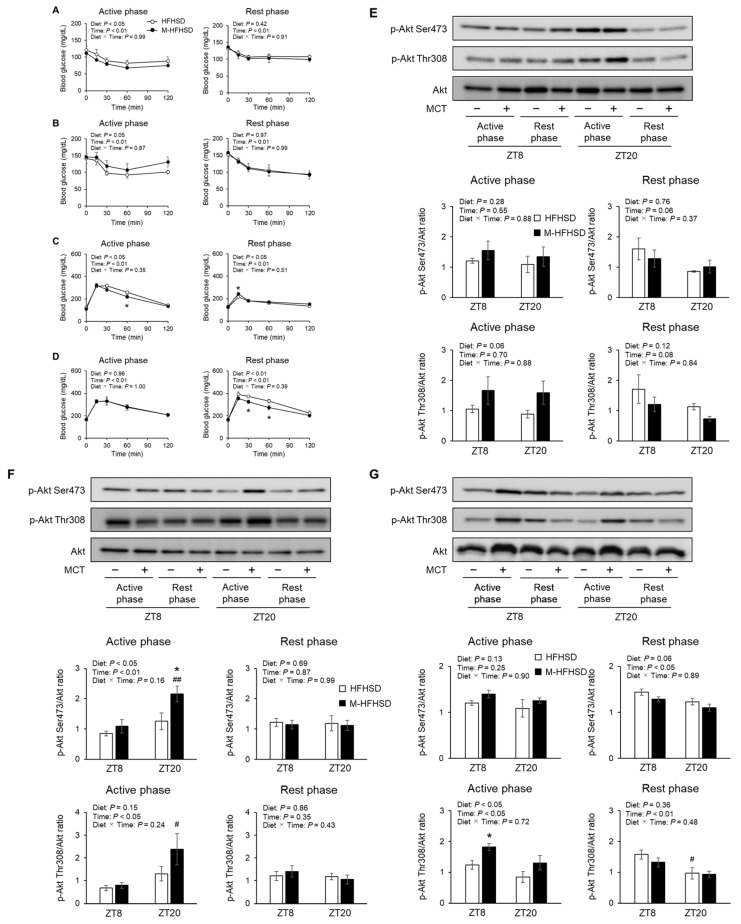
Insulin and glucose tolerance tests and insulin signaling in peripheral tissues. Insulin tolerance tests at feeding period (**A**) or fasting period (**B**) in mice fed HFHSD or M-HFHSD for 8 weeks. Glucose tolerance test at feeding period (**C**) or fasting period (**D**) in mice fed HFHSD or M-HFSHD for 8 weeks. Protein expression levels of phosphorylated Akt on Ser473 or Thr307 in the liver (**E**), gastrocnemius muscle (**F**), and epididymal white adipose tissue (**G**) extracts from mice. Data are mean ± SEM (*n* = 4). * *p* < 0.05 vs. HFHSD at same time. # *p* < 0.05 and ## *p* < 0.01 vs. same diet at ZT8. HFHSD, high-fat high-sucrose diet; MCT, medium-chain triglyceride; M-HFHSD, high-fat high-sucrose diet containing medium-chain triglyceride; ZT, zeitgeber time.

**Figure 5 nutrients-14-05096-f005:**
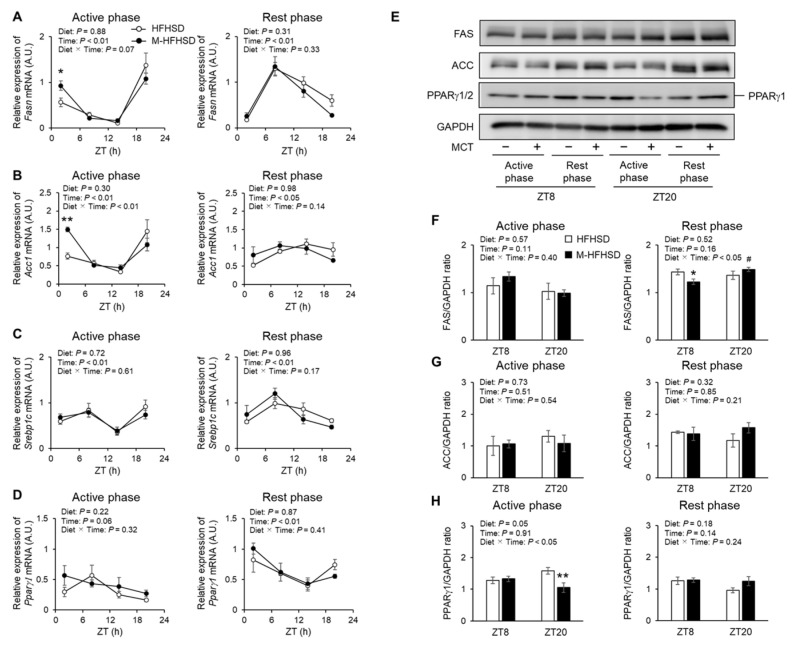
Expression of lipogenic genes and proteins in the liver. Expression levels of *Fasn* (**A**), *Acc1* (**B**), *Srebp1c* (**C**), and *Pparg1* (**D**) mRNA at indicated times in the liver of mice. (**E**) Protein expression levels of FAS, ACC, PPARγ1 and GAPDH in liver extracts from mice. (**F**–**H**) Ratios of FAS-, ACC-, and PPARγ1-to-GAPDH. Data are mean ± SEM (*n* = 4). * *p* < 0.05 and ** *p* < 0.01 vs. HFHSD at same time. # *p* < 0.05 vs. same diet at ZT8. HFHSD, high-fat high-sucrose diet; MCT, medium-chain triglyceride; M-HFHSD, high-fat high-sucrose diet containing medium-chain triglyceride; ZT, zeitgeber time.

**Figure 6 nutrients-14-05096-f006:**
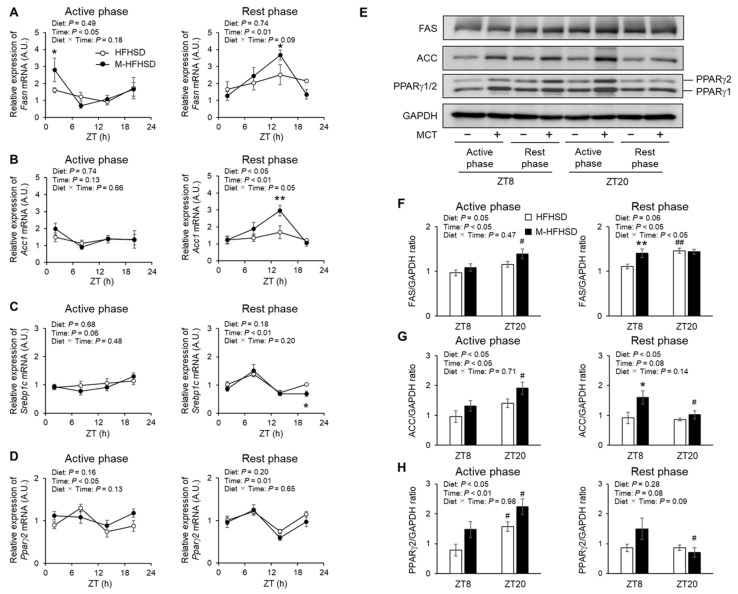
Expression of lipogenic genes and proteins in epididymal WAT. Expression levels of *Fasn* (**A**), *Acc1* (**B**), *Srebp1c* (**C**), and *Pparγ2* (**D**) mRNA at indicated times in epididymal WATs of mice. (**E**) Protein expression levels of FAS, ACC, PPARγ1/2, and GAPDH in eWAT extracts from mice. (**F**–**H**) Ratios of FAS-, ACC-, and PPARγ2-to-GAPDH. Data are mean ± SEM (*n* = 4). * *p* < 0.05 and ** *p* < 0.01 vs. HFHSD at same time. # *p* < 0.05 and ## *p* < 0.01 vs. same diet at ZT8. WAT, white adipose tissue; eWAT, epididymal white adipose tissue; HFHSD, high-fat high-sucrose diet; MCT, medium-chain triglyceride; M-HFHSD, high-fat high-sucrose diet containing medium-chain triglyceride; ZT, zeitgeber time.

**Figure 7 nutrients-14-05096-f007:**
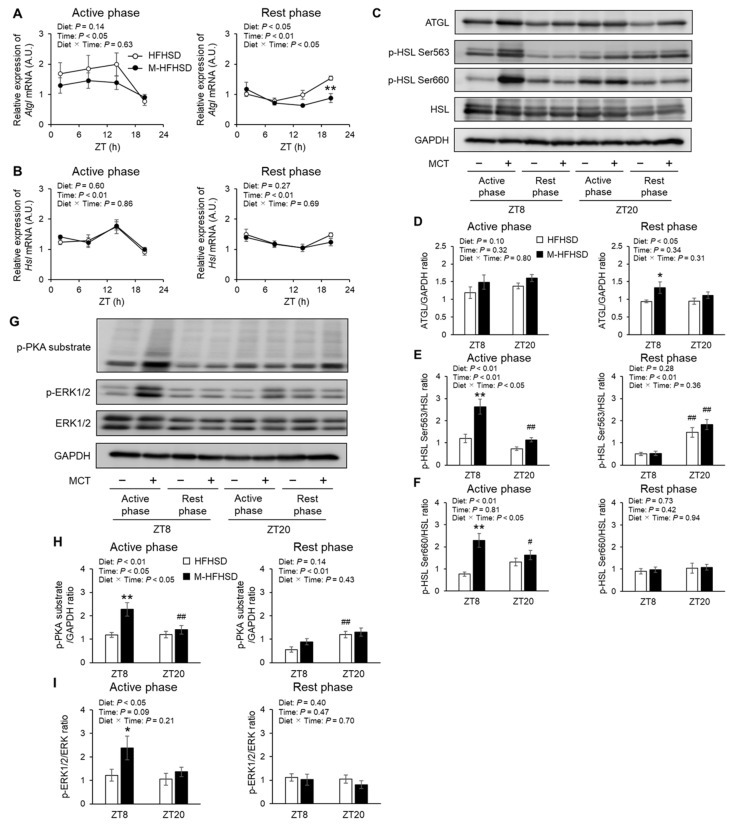
Expression of lipolytic genes and proteins and lipolysis-related signaling in epididymal white adipose tissue. Expression levels of *Atgl* (**A**) and *Hsl* (**B**) mRNA at indicated times in epididymal white adipose tissues of mice. (**C**) Protein expression of ATGL, phosphorylated HSL on Ser563, phosphorylated HSL on Ser660, total HSL and GAPDH in epididymal white adipose tissue extracts from mice. (**D**–**F**) Ratios of ATGL-to-GAPDH and phosphorylated protein-to-total protein. (**G**) Protein expression of phosphorylated PKA substrate, phosphorylated ERK, total ERK, and GAPDH in epididymal white adipose tissue extracts from mice. (**H**,**I**) Ratios of phosphorylated PKA substrate-to-GAPDH and phosphorylated ERK-to-total ERK. Data are mean ± SEM (*n* = 4). * *p* < 0.05 and ** *p* < 0.01 vs. HFHSD at same time. # *p* < 0.05 and ## *p* < 0.01 vs. same diet at ZT8. HFHSD, high-fat high-sucrose diet; MCT, medium-chain triglyceride; M-HFHSD, high-fat high-sucrose diet containing medium-chain triglyceride; ZT, zeitgeber time.

**Table 1 nutrients-14-05096-t001:** Composition of the diets.

Ingredients (g/100 g)	HFHSD	M-HFHSD
Casein	25	25
Cystine	0.375	0.375
Cornstarch	14.869	14.869
Sucrose	20	20
Beef tallow	14	7
Lard	14	8
MCT oil	0	13
Soybean oil	2	2
Cellulose	5	5
Mineral mixture, AIN-93	3.5	3.5
Vitamin mixture, AIN-93	1	1
Choline bitartrate	0.25	0.25
Tert-butyl hydroquinone	0.006	0.006
Fat (% of energy)	53.9	53.9
Total energy (MJ/100 g diet)	1.99	1.99

HFHSD, high-fat high-sucrose diet; MCT, medium-chain triglyceride; M-HFHSD and MCT-containing HFHSD.

## Data Availability

Not applicable.

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
