# Peer review of "Timing of Medium-Chain Triglyceride Consumption Modulates Effects in Mice with Obesity Induced by a High-Fat High-Sucrose Diet"

_nutrients, 2022, doi:10.3390/nu14235096_

Round 1
Reviewer 1 Report
In this manuscript (MS) entitled “Timing of medium-chain triglyceride consumption modulates effects in mice with obesity induced by a high-fat high-sucrose diet” the authors investigate that timing of MCT intake modulates glucose and lipid metabolism in mice fed high-fat and high-sucrose diet. MCT intake during the active phase reduces tissue weight gain in epididymal and subcutaneous fat. In addition, MCT intake during the active phase also increases insulin sensitivity. Therefore, the authors' results are important findings when considering the regulation of lipid and glucose metabolism by the timing of dietary intake. However, several points exist that may be confusing to the reader, so my comments are listed below. I encourage the authors to read my comments below.
Comments
1. Why is the number of samples not consistent in Figures 1 (n=6) and 2 (n=8) and subsequent figures?
2. Line 233-234, the authors describe that medium-chain triglyceride consumption led to increased insulin sensitivity in mice fed with HFHSD during the active, but not the rest phase. However, Figure 4A is missing a symbol indicating a significant difference. Were there significant differences at all times? The authors should indicate the symbols exactly at all times when there is a significant difference.
3. In Figure 4E, 4F and 5E, the quantitation does not reflect the actual band intensities. For example, western blotting show that p-Akt(Thr308) in ZT20 is increased MCT intake during the rest phase. However, quantitation of band intensities is not statistically different. The western blot data is out of sync with the data in the graph. Was the Western blot sample created by mixing the entire group or a randomly selected sample? If randomly selected, each group must have at least two samples.
4. In Figure 7H, the authors must analyze that total PKA levels in western blotting. It is inconsistent that only p-PKA is normalized by GAPDH, while p-ERK and p-HSL are normalized by total ERK and HSL, respectively.
5. In Line346-349, it is unclear why the authors are looking at phosphorylation levels of PKA and ERK. The authors must describe that HSL is phosphorylated by PKA and ERK using by references. In addition, regarding the assessment of Akt phosphorylation levels, it would be more helpful to the reader if you explained how glucose uptake is enhanced when Akt is phosphorylated.
6. MCT intake does not change the liver weight in Figure 2C. Nevertheless, the authors analyze the expression levels of lipogenic genes in the liver (Figure 3). Why did they analyze gene expression in liver with no weight variation?
7. In line 185-186, it is unclear what the authors mean by the statement in line. Is there an increase in body weight both in ad libitum and in the active phase of intake?
8. It is unclear what percentage of MCT are mixed in the feed. And, what is the control group mixing instead of MCT? The authors should describe these in the Materials and Methods section.
9. Line 402-403, the authors described that capric acid (C10:0) binds and partially activates 402 PPARγ, which induces the expression of FAS and ACC without inducing adipogenesis in 403 3T3-L1 cells [22,23]. However, in the ref. 22 and 23, there is no data of the effect of capric acid on ACC and FAS expression levels, which misleads the reader. Furthermore, isn't SREBP1 a major transcription factor that regulates the expression of lipogenic enzymes? (PMID: 10585467).
Author Response
Thank you for your constructive input.
Please refer to my responses to your valuable comments in the attachment.

Reviewer 2 Report
The author presents a comprehensive study about the diurnal effects of different diets and I have only minor comments.
Unfortunately, the supplement data and tables are missing for evaluation. When provided I will check again.
You write "we" sometimes, but you are the only author and no acknowledgement is included. Do you need to name other authors?
Line 182: Isn't it the other way around? Mice with M-HFHSD weighted less in your diagram?
Lines 275-279: Names of genes in the text need to be written italic. The name of the gene fatty acid synthase is Fasn.
Author Response
Thank you for your time and your positive comment.
Please refer to our responses to your valuable comments in the attachment.

Round 2
Reviewer 1 Report
The author have made substantial efforts and included additional data to ansewer questions and comments made by the reviewer. I am satisfied with the revisions that have been made by the author. However, I would have liked a letter describing the responses to the comments.
Reviewer 2 Report
Thanks for the corrections!
Line 148: One wrong colon
You are not referring to Supplementary Fig 1C and Supplementary Fig 4 in the text.